# Hemodialysis Intensifies NLRP3 Inflammasome Expression and Oxidative Stress in Patients with Chronic Kidney Disease

**DOI:** 10.3390/ijms26146933

**Published:** 2025-07-19

**Authors:** Marcia Ribeiro, Ludmila F. M. F. Cardozo, Karen Salve Coutinho-Wolino, Marcelo Ribeiro-Alves, Denise Mafra

**Affiliations:** 1Graduate Program in Biological Sciences—Physiology, Carlos Chagas Filho Institute of Biophysics (IBCCF), Federal University of Rio de Janeiro (UFRJ), Rio de Janeiro 21941-902, RJ, Brazil; marciaribeiro@biof.ufrj.br (M.R.); karenscoutinho@gmail.com (K.S.C.-W.); 2Graduate Program in Nutrition Sciences, Fluminense Federal University (UFF), Niterói 24020-140, RJ, Brazil; ludmilacardozo@id.uff.br; 3Graduate Program in Cardiovascular Sciences, Fluminense Federal University (UFF), Niterói 24070-090, RJ, Brazil; 4HIV/AIDS Clinical Research Center, National Institute of Infectology (INI/Fiocruz), Rio de Janeiro 21040-360, RJ, Brazil; mribalves@gmail.com

**Keywords:** NLRP3, inflammasome, chronic kidney disease, hemodialysis, peritoneal dialysis

## Abstract

Chronic inflammation plays a central role in the progression and complications of chronic kidney disease (CKD). The nod-like receptor pyrin domain-containing 3 (NLRP3) inflammasome pathway has emerged as a crucial mediator of the inflammatory response in CKD. This cross-sectional study evaluated the expression of NLRP3 in patients with CKD undergoing different treatments. Blood samples were collected from 32 non-dialysis (ND) patients [63 (11.2) years, estimated glomerular filtration rate, 43.5 (22.0) mL/min, BMI, 29.5 (10.0) kg/m^2^)], 50 hemodialysis (HD) patients [48.5 (16.5) years, 60.5 (50) months on HD, BMI, 24.2 (4.9) kg/m^2^)], and 8 peritoneal dialysis (PD) patients [56.5 (8.5) years, 40.5 (41.2) months on PD, BMI, 28.8 (2.6) kg/m^2^)]. The mRNA expression level of NLRP3 was measured using real-time PCR. The cytokines and the malondialdehyde (MDA) levels were also assessed. The results indicated that the mRNA level of NLRP3 was significantly elevated in patients undergoing HD (1.23, IQR = 0.95) compared with that in non-dialysis patients (0.79, IQR = 0.35) and in patients undergoing PD (0.77, IQR = 0.38) after adjusting for confounding variables, including age, sex, BMI, and dialysis duration. Furthermore, the MDA levels were significantly higher in HD patients. NLRP3 is upregulated in HD patients, and the results suggested that the inflammasome may be associated with oxidative stress in patients with CKD.

## 1. Introduction

Inflammasomes are supramolecular complexes in the cytosol that activate inflammatory caspases. The nod-like receptor pyrin domain-containing (NLRP3) inflammasome is the most widely studied in the literature [1]. Also known as cryopyrin, NLRP3 belongs to a family of proteins characterized by a nucleotide-binding domain (NBD) and leucine-rich repeats (LRRs). It is described as a sensor of cellular stress and plasma membrane damage. It plays a central role in classical pyroptosis, triggering the activation of pro-caspase-1, the cleavage of gasdermin D (GSDMD), and the release of pro-inflammatory cytokines, promoting inflammation and cell death [1,2].

The NLRP3 inflammasome is activated in two steps: the priming phase and the activation phase. Priming of NLRP3 can occur through pattern recognition receptors (PRRs) via the detection of pathogen-associated molecular patterns (PAMPs) or damage-associated molecular patterns (DAMPs) [3,4,5]. Additionally, NLRP3 expression can be initiated by cytokines involved in immune responses and inflammatory processes, such as NF-κB activation or other transcription factors, leading to an upregulation of the inflammasome and its components [3,6]. NLRP3 is triggered by various exogenous microorganisms or endogenous molecules that commonly lead to K^+^ efflux or other ionic disturbances [3]. After activation, NLRP3 recruits downstream components to form the inflammasome complex, including the adaptor protein apoptosis-associated speck-like protein containing a caspase-recruitment domain (ASC). This protein facilitates the recruitment of pro-caspase-1, which is activated through autocatalytic cleavage, resulting in the generation of active caspase-1. This process promotes the maturation of the pro-inflammatory cytokines IL-1β and IL-18, triggers immune responses, and induces pyroptosis—a form of inflammatory cell death [1,2,3,6].

NLRP3 is present in various cell types, including neutrophils, monocytes, lymphocytes, and even neurons. Its activation has been associated with several human diseases, including metabolic disorders such as type 2 diabetes and obesity; central nervous system diseases like Alzheimer’s and Parkinson’s; and various types of cancer, and it is also involved in the pathophysiology of chronic kidney disease (CKD) [3,6].

CKD represents a progressive and irreversible condition that affects approximately 700–840 million people worldwide. It is characterized by abnormalities in the structure or function of the kidneys that persist for more than 3 months. CKD is divided into five stages depending on the glomerular filtration rate (GFR). Conservative treatment (non-dialysis) can be implemented in stages 1 to 5 (without dialysis). Renal replacement therapy (RRT) usually occurs in stage 5 and can be hemodialysis, peritoneal dialysis, or a kidney transplant [7].

Patients with CKD commonly develop complications such as hyperkalemia, metabolic acidosis, hyperphosphatemia, anemia, vitamin D deficiency, and secondary hyperparathyroidism [8]. In addition to these alterations, oxidative stress, immune dysfunction, and low-grade chronic inflammation are prevalent, as evidenced by elevated circulating inflammatory markers, including IL-1β, IL-6, and C-reactive protein [9]. Furthermore, NLRP3 levels and its components are upregulated in patients with CKD, thereby amplifying the inflammatory response. Indeed, Zhang et al. (2020) [10] observed that peripheral blood mononuclear cells from patients with CKD undergoing hemodialysis exhibited higher NLRP3 mRNA expression than those from healthy individuals. However, no study to date has evaluated whether there are differences in NLRP3 levels across the various stages and treatments of CKD. Therefore, this study aimed to assess the mRNA expression of NLRP3 in peripheral blood mononuclear cells (PBMCs) in CKD patients undergoing non-dialysis (ND), hemodialysis (HD), and peritoneal dialysis (PD).

## 2. Results

In total, 90 patients with CKD were analyzed: 32 patients with CKD stage III to V, 50 HD patients, and 8 PD patients. Table 1 presents the general characteristics of the patients in the three groups. Regarding the characterization of the sample, based on an analysis of the medical records, a predominance of arterial hypertension was observed among the evaluated patients: 96% of patients were diagnosed with systemic arterial hypertension in the ND group, 82% in the HD group, and 79% in the DP group. Additionally, 36% had type II diabetes mellitus in ND, 19% in HD, and 30% in the DP group.

Figure 1 shows boxplots comparing the relative mRNA expression of NLRP3, Nrf2, and NF-κB among non-dialysis (ND), hemodialysis (HD), and peritoneal dialysis (PD) patients. Patients undergoing HD exhibited significantly higher NLRP3 mRNA expression compared with those in both ND (*p* < 0.0001) and PD (*p* = 0.01). This finding supports enhanced inflammasome activation in the HD group. No significant differences in Nrf2 expression were observed among the groups. This suggests that antioxidant defense pathways mediated by Nrf2 are not differentially regulated across dialysis modalities. Similarly, NF-κB expression did not differ significantly between groups, despite the increased NLRP3 expression in HD patients. This may indicate that the inflammatory activation in HD involves mechanisms beyond NF-κB transcriptional regulation.

Figure 2 shows that patients undergoing hemodialysis (HD) exhibited significantly higher MDA plasma levels than those not on dialysis (ND) and those on peritoneal dialysis (PD). Regarding the circulating levels of TNF-α, patients undergoing PD exhibited significantly higher plasma levels than those in both ND (*p* < 0.03) and HD (*p* = 0.0001). The IL-6 plasma levels showed a tendency to be high in patients undergoing PD (*p* = 0.08).

This scatter plot (Figure 3) shows a positive correlation between NLRP3 mRNA expression and MDA plasma levels among patients. The Pearson correlation coefficient was r = 0.26 with a *p*-value of 0.01, indicating a statistically significant but modest association. These results suggest that higher inflammasome gene expression (NLRP3) may be associated with increased lipid peroxidation, as indicated by elevated MDA levels.

## 3. Discussion

This cross-sectional study of patients with chronic kidney disease (CKD) reveals that NLRP3 mRNA expression is higher in patients undergoing hemodialysis (HD) than in PD and non-dialysis CKD patients. Additionally, NLRP3 mRNA expression was positively correlated with plasma malondialdehyde (MDA) levels, suggesting that lipid peroxidation may be involved in the NLRP3 inflammasome in patients with CKD.

As previously mentioned, the NLRP3 inflammasome is a cytosolic complex formed by proteins of the NBD-LRR family that detects cellular stress and damage to the plasma membrane. Its activation is triggered by several factors, including high glucose, fatty acids, and oxidative stress, and involves two steps [1,9]. In the first phase, the priming phase, pattern recognition receptors detect PAMPs or DAMPs, or through cytokines that activate transcription factors such as NF-κB, leading to increased NLRP3 expression. The second phase comprises the activation phase; stimuli such as K^+^ efflux or ionic disturbances induce its assembly [3,4,5,6].

NLRP3 recruits the adaptor protein ASC and pro-caspase-1, which is activated by autocleavage, resulting in the generation of active caspase-1. This, in turn, promotes the maturation of the inflammatory cytokines IL-1β and IL-18 and induces pyroptosis, a form of inflammatory cell death. NLRP3 activation is associated with diseases such as type 2 diabetes, obesity, Alzheimer’s, Parkinson’s, cancers, and CKD [3,4,5,6].

In chronic kidney disease, mitochondrial dysfunction, oxidative stress, immune dysfunction, and chronic low-grade inflammation are common, as reflected by elevated levels of circulating inflammatory markers, such as IL-1β, IL-6, and C-reactive protein [9]. Corroborating this, there is also a higher expression of NLRP3 and its components in CKD, which amplifies the inflammatory response, promoting pyroptotic cell death and a consequent clinical worsening of CKD [1,9]. It has been reported in the literature that patients undergoing HD have overexpression of nuclear factor kappa-B (NF-κB) compared with healthy individuals. In contrast, non-dialysis CKD patients may maintain a regular balance between the expressions of NF-κB and nuclear factor erythroid 2-related factor 2 (Nrf2) [11]. During the NLRP3 activation phase, TNF or Toll-like receptors stimulate NF-κB, which in turn enhances the expression of NLRP3 and IL-1β proteins [9]. Although our study did not observe correlations between NLRP3 and NF-κB, the overexpression of NF-κB may contribute to the activation and upregulation of NLRP3. In line with our findings, Leal et al. (2015) [11] reported increased NF-κB expression in patients undergoing HD compared with those with non-dialysis, which may help explain the elevated NLRP3 levels observed in our study.

Thus, it is clear that the persistent low-grade inflammation observed in CKD from the early stages can lead to the activation and expression of NLRP3, which in turn will further amplify inflammation in CKD. Hashemi et al. (2021) [12] evaluated the activation of the NLRP3 inflammasome in peripheral blood mononuclear cells from 28 patients with glomerulonephritis treated with hemodialysis. They observed that, compared with PBMCs from healthy individuals, IL-1β and IL-18, NLRP3, ASC, and CASP-1 were markedly overexpressed in those patients. The authors state that these results revealed that the NLRP3–ASC–caspase-1 axis may play a role in the increased severity of inflammation reported in these patients, in addition to providing new insights into the molecular mechanisms underlying chronic inflammation in patients with glomerulonephritis-HD [12].

Studies have shown that the activation of the endothelial NLRP3 inflammasome contributes to peritoneal inflammation and fibrosis in patients with CKD on PD and is also associated with defects in solute transport and tissue remodeling during PD-related peritonitis [13,14]. However, patients undergoing HD are more exposed to pro-inflammatory stimuli, including direct contact with the artificial surfaces of dialyzers and artificial tubes, complement microactivation, the release of pro-inflammatory cytokines, and the production of reactive oxygen species (ROS) [15]. This was demonstrated in a study by Sahib et al. (2024) [16], who evaluated the inflammatory state in patients with CKD undergoing HD compared with those on PD and observed that the inflammatory burden is greater in patients undergoing HD. In PD, which is performed continuously and more physiologically, the exposure is lower, leading to less activation of the immune system through contact with foreign surfaces. This results in reduced levels of systemic inflammation and, consequently, less activation of the NLRP3 inflammasome [15,16].

The literature has already described the bidirectional relationship between NLRP3 inflammasome activation and oxidative stress. The overproduction of ROS in patients with CKD may be a part of the vast array of inflammasome-activating signals. On the other hand, the NLRP3 inflammasome activation induces ROS-producing processes [17]. High levels of MDA may indicate an impaired ability to neutralize ROS derived from oxygen and lipids, resulting in the formation of lipid peroxides. While a direct link between MDA and NLRP3 has yet to be established, oxidative stress (including MDA as a lipid peroxidation) may modulate NLRP3 activation through its regulatory effects on NF-κB signaling [18,19].

Moreover, this study demonstrated that patients undergoing HD had higher MDA levels than ND and PD patients. This finding is consistent with the literature. For instance, Capusa et al. [20] also showed that plasma levels of thiobarbituric acid-reactive substances (TBARSs), a by-product of MDA, were significantly higher in HD patients than in non-dialyzed and PD patients. Another study indicated that MDA levels were significantly higher after HD sessions in patients with end-stage renal failure [21], reinforcing the findings that lipid peroxidation may be increased in HD patients.

Indeed, several factors contribute to the accumulation of oxidized products, such as MDA, in these patients, including the loss of antioxidants during the HD session, the presence of a central venous catheter, and the use of bioincompatible dialyzers, in addition to other factors related to the kidney disease itself [22]. Regarding MDA, ROS are the primary source of lipid peroxidation. One method of blocking this process involves activating antioxidant enzymes, such as superoxide dismutase and glutathione peroxidase. However, in CKD, particularly in HD, these enzymes are diminished, suggesting that the increase in lipid peroxidation in these patients is likely linked to an inefficient antioxidant defense [23]. Interestingly, although HD and PD are renal replacement therapies, MDA levels vary based on the type of dialysis. Consequently, MDA is lower in the leukocytes of PD patients than in those undergoing HD. This difference arises because PD appears to be more associated with protein oxidation, whereas HD seems more connected to lipid peroxidation [24], supporting the data of this study. The reason for this phenomenon remains unclear; therefore, future studies are encouraged to investigate the oxidized products, particularly proteins and lipids, in various types of dialysis in more detail.

Our study is a pioneer in evaluating and comparing NLRP3 mRNA expression in peripheral blood mononuclear cells in CKD patients undergoing non-dialysis (ND), hemodialysis (HD), and peritoneal dialysis (PD). Our findings indicate that HD patients exhibit elevated inflammasome activation compared with patients with ND and PD. Furthermore, lipid peroxidation may be associated with the activation of the NLRP3 inflammasome. Thus, the NLRP3 inflammasome pathway may be a promising therapeutic target for preventing complications in CKD patients, especially those undergoing HD.

### Limitations

Although the sample size was limited, appropriate statistical analyses were used to minimize the effects of non-normal distributions and outliers. However, the statistically significant differences observed should be interpreted carefully. Additionally, although conservative multiple-comparison adjustments (Tukey–Kramer HSD) were applied, the small sample size in the peritoneal dialysis group may result in unstable estimates. Therefore, interpretation should be cautious. Although a weak correlation was observed between NLRP3 mRNA expression and MDA levels, this association explained less than 7% of the variance, highlighting the limited interpretability of this finding. Given the modest sample size and cross-sectional design, these results should be interpreted with caution and considered exploratory.

Replication with larger cohorts and the use of bootstrapping methods to estimate confidence intervals could enhance the reliability of these findings and mitigate the impact of sample size variability. Future studies should include larger samples.

## 4. Materials and Methods

### 4.1. Study Design and Patients

This is a cross-sectional analysis of baseline data from non-dialysis CKD patients (stages 3–5) and patients undergoing renal replacement therapy, including hemodialysis (three dialysis sessions per week, each lasting 4 h with arteriovenous fistula as vascular access) and continuous ambulatory peritoneal dialysis (CAPD). These baseline data were derived from cohorts previously published by our group [25,26]. Patients aged 18–75 years with CKD stages III and V who were treated at the renal nutrition outpatient clinic of Fluminense Federal University and received nutritional monitoring with a low-protein diet for at least six months (0.6 g/kg/day) were included. Dialysis patients, including men and women aged 18 to 75 years, were recruited from dialysis centers, provided they had received at least 6 months of dialysis treatment (hemodialysis or CAPD).

Pregnant women; transplant patients; patients with liver, autoimmune or infectious diseases; patients with cancer, heart failure, or acquired immunodeficiency syndrome; patients using catabolic drugs; and those who were using antibiotics and anti-inflammatory medications up to three months before the start of the study were not included. This study is embedded within two research protocols previously approved by the Faculty of Medicine Ethics Committee at Fluminense Federal University, under the approval numbers 4.449.525 and 4.209.791. Weight and height were collected to calculate body mass index (BMI).

### 4.2. Analytic Procedures and Sample Processing

Blood samples were collected from each patient after an overnight fast using a syringe containing EDTA (1 mg/mL) as an anticoagulant. Blood samples from hemodialysis patients were collected immediately before the start of the dialysis session, at the exact time of venous access puncture. Plasma was separated (15 min, 3800 rpm, 4 °C) and stored at −80 °C until analysis.

Plasma levels of cytokines such as interleukin 6 (IL-6) and tumor necrosis factor-alpha (TNF-α) were determined using an enzyme-linked immunosorbent assay (ELISA) (PeproTech EC Ltd., London, United Kingdom). Plasma malondialdehyde (MDA) levels were obtained by lipid peroxidation analysis [27]. Dialysis time was calculated from data obtained from the medical records. The estimated glomerular filtration rate (eGFR) was calculated using the Chronic Kidney Disease Epidemiology Collaboration (CKD-EPI) formula, as recommended by the Clinical Practice Guidelines for Kidney Disease Improving Global Outcomes (KDIGO) [28].

EDTA blood samples were diluted in PBS to obtain peripheral blood mononuclear cells (PBMCs). The PBMCs were collected and washed twice with cold PBS and then resuspended and stored at −80 °C in 1 mL of frozen cell culture recovery medium (Gibco, Thermofisher^®^, Waltham, MA, USA) for RNA isolation. PBMCs were isolated from the blood, and RNA was extracted using an SV Total RNA Isolation System kit (Promega^®^, Madison, WI, USA). The cDNA was synthesized with the high-capacity cDNA reverse transcription kit (Thermofisher^®^, Waltham, MA, USA).

### 4.3. Quantitative Real-Time PCR Analyses

NLRP3, NRF2, and NF-kB mRNA levels were assessed in PBMCs using quantitative real-time polymerase chain reaction (RT-qPCR). RT-qPCR amplification reactions were performed in duplicate in 20 µL of final volume via TaqMan Gene Expression Assays on the ABI Prism 7500 Sequence Detection System (Applied Biosystems^®^, Waltham, MA, USA). The PCR protocol was performed using TaqMan Primer Assays (Thermofisher^®^, Waltham, MA, USA) for NLRP3 (Hs00918082_m1), NRF2 (Hs00975961_g1), NF-kB (Hs00765730_m1), and the control gene, GAPDH (Hs02758991_g1): 50 °C for 2 min, 95 °C for 10 min, and 40 two-step cycles: 95 °C for 15 s and 60 °C for 1 min. The NLRP3, NRF2, and NF-kB mRNA expressions were normalized against those of GAPDH, and the expression levels were calculated using the 2^−ΔΔTC^ (delta–delta threshold cycle) method.

### 4.4. Statistical Analyses

Data is presented as medians with interquartile ranges (IQRs), representing the spread between the 75th and 25th percentiles, or as frequencies (percentages) for categorical data, and the comparisons among the groups (i.e., in non-dialysis, PD, and HD CKD patients) were tested by either Kruskal–Wallis (continuous-numerical variables) or chi-squared (discrete-nominal variables) tests. Non-parametric methods were employed to address the unequal group sizes. All models were adjusted for confounding variables (i.e., age, sex, BMI, and dialysis duration) where applicable. All other variables in the linear multiple fixed-effect models were held at their mean values or at equal proportions to estimate marginal mean values for the groups. Contrasts were constructed from these estimated mean marginal effects. The Tukey Honest Significant Difference (HSD) method was used to correct *p*-values by the number of comparisons. Similarly, correlation analyses were conducted using Pearson’s coefficients after adjustments for the confounding variables. Statistical significance was determined at *p* < 0.05, and all analyses were conducted using R version 4.2.1.

## 5. Conclusions

The NLRP3 mRNA expression is upregulated in patients with CKD undergoing HD compared with that in non-dialysis patients and those receiving PD. Additionally, the results suggest that the inflammasome may be linked to oxidative stress in these patients.

## Figures and Tables

**Figure 1 ijms-26-06933-f001:**
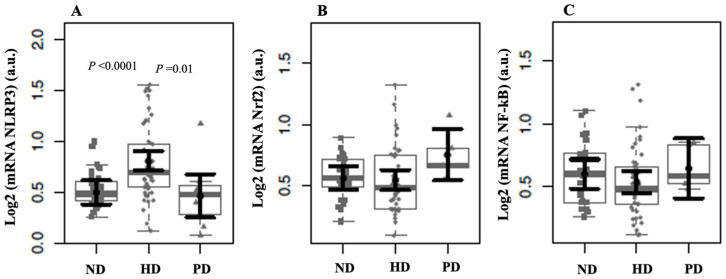
NLRP3 (**A**), Nrf2 (**B**), and NF-kB (**C**) mRNA expressions in non-dialysis (ND) patients, hemodialysis (HD), and peritoneal dialysis (PD) patients. The data distributions are represented in box and strip plots. In black, the center circles represent the mean marginal effects for each group estimated from a linear fixed-effects model adjusted for confounding variables (i.e., age and sex) by holding all other variables in the linear multiple fixed-effect models at their mean values or equal proportions.

**Figure 2 ijms-26-06933-f002:**
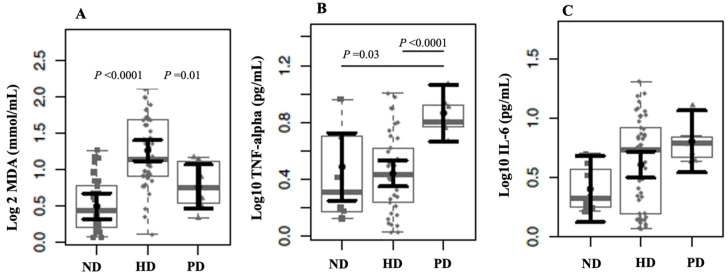
MDA (**A**), TNF-α (**B**), and IL-6 (**C**) plasma levels in non-dialysis (ND) patients, hemodialysis (HD), and peritoneal dialysis (PD) patients. The data distributions are represented in box and strip plots. In black, the center circles represent the mean marginal effects for each group estimated from a linear fixed-effects model adjusted for confounding variables (i.e., age and sex) by holding all other variables in the linear multiple fixed-effect models at their mean values or equal proportions.

**Figure 3 ijms-26-06933-f003:**
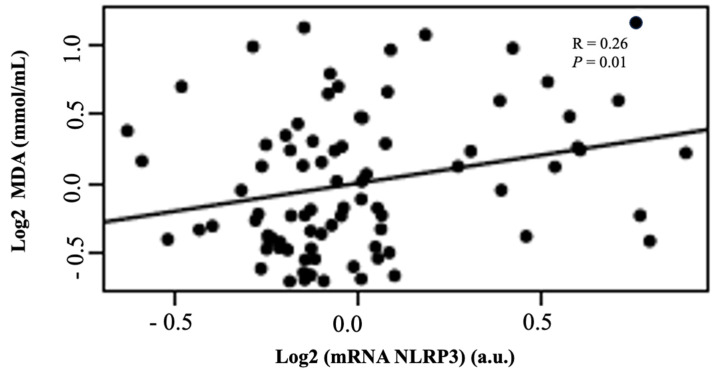
Correlation between malondialdehyde (MDA) plasma levels and NLRP3 mRNA expressions in patients with CKD, adjusted for age, sex, BMI, and dialysis duration.

**Table 1 ijms-26-06933-t001:** General characteristics of ND, HD, and PD CKD patients.

Parameters	ND (N = 32)	HD (N = 50)	PD (N = 8)	*p* Values
Sex (male/female)	14/18	20/30	5/3	0.490
Age (years)	63 (11.2)	48.5 (16.5)	56.5 (8.5)	0.000
Time on dialysis (months)	-	60.5 (50)	40.5 (41.2)	0.187
eGFR (mL/min/1.72 m^2^)	43.5 (22.0)	-	-	-
BMI (kg/m^2^)	29.5 (10.0)	24.2 (4.9)	28.8 (2.6)	0.003

Data expressed as median and interquartile range (IQR) or absolute and relative. *p* values estimated by nonparametric Kruskal–Wallis (continuous numerical variables). Abbreviations: ND: non-dialysis; HD: hemodialysis; PD: peritoneal dialysis; eGFR: estimated glomerular filtration rate; BMI: body mass index.

## Data Availability

The datasets used and/or analyzed during the current study are available from the corresponding author upon reasonable request.

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
