# Peer review of "Hemodialysis Intensifies NLRP3 Inflammasome Expression and Oxidative Stress in Patients with Chronic Kidney Disease"

_ijms, 2025, doi:10.3390/ijms26146933_

Round 1

Reviewer 1 Report

Comments and Suggestions for Authors

The aim of this cross-sectional study is to evaluate NLRP3 mRNA expression in peripheral blood mononuclear cells of patients with CKD, stratified by treatment modality: non-dialysis, hemodialysis, and peritoneal dialysis. The study also investigates the association between NLRP3 expression and oxidative stress, measured via MDA plasma levels. The topic is relevant and timely, given the increasing interest in the role of innate immunity and oxidative stress in CKD. Major Limitations:

Critical lack of clinical characterization of patients: The absence of essential data on comorbidities such as diabetes, cardiovascular disease, neoplasia, autoimmune conditions, and infectious diseases (beyond the basic exclusion criteria) is a major flaw. These factors are known to profoundly influence both oxidative stress and NLRP3 activation, and their omission severely compromises the interpretability of the results. In particular, the observed increase in NLRP3 expression in HD patients could be entirely or partially attributable to underlying conditions (e.g., diabetes, cardiovascular inflammation, infections), rather than to dialysis per se.

Inadequate analysis of graphical data: The presentation and interpretation of the figures are insufficiently described.

The PD group is too small (n = 8) to allow meaningful comparisons. The limited statistical power in this group needs to be acknowledged and the conclusions toned down accordingly.

The cross-sectional design precludes causal inference between dialysis modality and NLRP3 expression.

Only mRNA expression was measured. No protein-level data (e.g., active caspase-1, IL-1β, IL-18) were provided to confirm functional inflammasome activation.

Critical Comments and Required Revisions- Major Revisions Required

Provide detailed clinical data on all patient groups:

At a minimum: prevalence of diabetes mellitus, cardiovascular disease (e.g., coronary artery disease, heart failure), malignancy, chronic infections, and autoimmune diseases.

Include relevant laboratory data if available: CRP, albumin, hemoglobin, lipid profile.

Specify dialysis-related variables: vascular access type (AVF vs CVC), dialyzer membrane (biocompatible vs non-biocompatible), use of HDF.

Re-analyze or clearly re-interpret the results in light of potential confounders, especially for the HD group. Without these data, the attribution of elevated NLRP3 expression to dialysis is speculative.

Improve all figure presentations

Reassess the correlation analysis (e.g., NLRP3 vs MDA): is it robust after multivariable adjustment?

Clarify whether blood sampling in HD patients was performed pre- or post-dialysis.

Author Response

Reviewer 1

Critical lack of clinical characterization of patients: The absence of essential data on comorbidities such as diabetes, cardiovascular disease, neoplasia, autoimmune conditions, and infectious diseases (beyond the basic exclusion criteria) is a major flaw. These factors are known to profoundly influence both oxidative stress and NLRP3 activation, and their omission severely compromises the interpretability of the results. In particular, the observed increase in NLRP3 expression in HD patients could be entirely or partially attributable to underlying conditions (e.g., diabetes, cardiovascular inflammation, infections), rather than to dialysis per se.

Response: Thank you for this comment. We agree that sample characterization is extremely important and reflects the results obtained. We have added sample characterization concerning the comorbidities of these patients to the results. Also, we have added the exclusion criteria. It was our mistake.

Inadequate analysis of graphical data: The presentation and interpretation of the figures are insufficiently described.

Response: Thank you. We have improved them.

The PD group is too small (n = 8) to allow meaningful comparisons. The limited statistical power in this group needs to be acknowledged and the conclusions toned down accordingly.

Response: Thank you for this critical comment. We fully agree that the peritoneal dialysis (PD) group represents a small sample size (n = 8), which inevitably limits statistical power. However, it is essential to note that this reflects the actual distribution of dialysis modalities in Brazil and globally. According to the Brazilian Society of Nephrology, only approximately 4.7% of dialysis patients are on PD. In comparison, hemodialysis accounts for over 90%. International data also confirm that the prevalence of PD is around 7% worldwide. Therefore, recruiting larger numbers of PD patients within a single-center study is highly challenging. Nonetheless, we acknowledge this limitation in our manuscript and agree that the conclusions regarding the PD group should be interpreted with caution. We have also emphasized this point in the discussion section. Additionally, we employed robust statistical methods to minimize the impact of the small sample size as much as possible

The cross-sectional design precludes causal inference between dialysis modality and NLRP3 expression.

Response: Thank you for this critical observation. We fully agree that the cross-sectional nature of our study limits any causal inference regarding the relationship between dialysis modality and NLRP3 expression. We have clarified this point in the Discussion section, emphasizing that our findings demonstrate associations rather than causality and should be interpreted accordingly.

Only mRNA expression was measured. No protein-level data (e.g., active caspase-1, IL-1β, IL-18) were provided to confirm functional inflammasome activation.

Response: We appreciate this comment. We agree that evaluating the other proteins involved in the inflammasome activation cascade would greatly enrich the study; however, we emphasize that we evaluated the gene expression of the inflammasome through a sophisticated rt-PCR analysis, which requires the use of high-cost primers, and for this reason, it was not possible to assess other markers. Furthermore, we emphasize that evaluating only the expression of the NLRP3 gene does not invalidate the relevance of our study, which aimed to explore the gene expression of the inflammasome in this specific context. To our knowledge, no previous studies have addressed this question in patients undergoing dialysis. Naturally, future research using larger samples and additional protein-level measurements will be essential to confirm and expand our findings.

Provide detailed clinical data on all patient groups:

Response: Thank you for this suggestion. We have added more detailed patient clinical information to the study methods. However, since these data were comprehensively reported in our previous publications (Reis et al., 2024; Chermut et al., 2023) we have opted not to repeat the same results in this manuscript to avoid redundancy.

At a minimum: prevalence of diabetes mellitus, cardiovascular disease (e.g., coronary artery disease, heart failure), malignancy, chronic infections, and autoimmune diseases.

Response: Thank you for this comment. Patients with malignancies, chronic infections, heart failure, cancer, and autoimmune diseases were excluded from the study and are now described in the methods. The prevalence of hypertensive and diabetic patients was also included in the results.

Include relevant laboratory data if available: CRP, albumin, hemoglobin, lipid profile.

Response: As we explain above, since these data were comprehensively reported in our previous publications (Reis et al., 2024; Chermut et al., 2023) we have opted not to repeat the same results in this manuscript to avoid redundancy.

Specify dialysis-related variables: vascular access type (AVF vs CVC), dialyzer membrane (biocompatible vs non-biocompatible), use of HDF.

Response: Thank you for your comment. The information has been added to the methods.

Re-analyze or clearly re-interpret the results in light of potential confounders, especially for the HD group. Without these data, the attribution of elevated NLRP3 expression to dialysis is speculative.

Response: Thank you for this comment. The data were statistically analyzed, taking into account the confounders such as age, BMI, time on dialysis, and sex.

Improve all figure presentations

Response: Thank you. We have improved them.

Reassess the correlation analysis (e.g., NLRP3 vs MDA): is it robust after multivariable adjustment?

Response: We appreciate the reviewer’s important point. We confirm that the correlation analysis between NLRP3 mRNA levels and plasma MDA was performed after adjusting for potential confounding factors, including age, sex, dialysis duration, and body mass index (BMI), using multivariable analysis. The correlation coefficient reported in the manuscript reflects this adjusted analysis. Although the correlation was modest (r=0.26), it remained statistically significant after these adjustments. We have clarified this in the Methods and Results sections to avoid any ambiguity.

Clarify whether blood sampling in HD patients was performed pre- or post-dialysis.

Response: Thank you for this comment. We clarified in the study methods that blood collection from hemodialysis patients was performed before the dialysis session, at the time of access puncture.

Once again, we thank the Reviewer for their constructive criticism, which has enabled us to improve the manuscript to the point where we hope it can be accepted for publication in IJMS.

Denise Mafra, on behalf of the rest of the co-authors.

References:

Chermut TR, Fonseca L, Figueiredo N, de Oliveira Leal V, Borges NA, Cardozo LF, Correa Leite PE, Alvarenga L, Regis B, Delgado A, Berretta AA, Ribeiro-Alves M, Mafra D. Effects of propolis on inflammation markers in patients undergoing hemodialysis: A randomized, double-blind controlled clinical trial. Complement Ther Clin Pract. 2023 May;51:101732. doi: 10.1016/j.ctcp.2023.101732. Epub 2023 Jan 26. PMID: 36708650.

 Reis DCMV, Alvarenga L, Cardozo LFMF, Baptista BG, Fanton S, Paiva BR, Ribeiro-Alves M, Fortunato RS, Vasconcelos AL, Nakao LS, Sanz CL, Berretta AA, Leite M Jr, Mafra D. Can curcumin supplementation break the vicious cycle of inflammation, oxidative stress, and uremia in patients undergoing peritoneal dialysis? Clin Nutr ESPEN. 2024 Feb;59:96-106. doi: 10.1016/j.clnesp.2023.11.015. Epub 2023 Nov 28. PMID: 38220413.

Reviewer 2 Report

Comments and Suggestions for Authors

Comments on the manuscript titled: Hemodialysis Intensifies NLRP3 Inflammasome Expression and Oxidative Stress in Patients with Chronic Kidney Disease by Marcia Ribeiro et al., submitted to IJMS mdpi Journal

This is a very valuable report based on experimental research.The study showed that patients with chronic kidney disease (CKD) undergoing haemodialysis have higher expression of the NLRP3 inflammasome compared to non-dialytic patients and those undergoing peritoneal dialysis. It was also identified that levels of oxidative stress markers, such as MDA, are higher in haemodialysis patients. The results suggest that haemodialysis may increase inflammation and oxidative stress, which contributes to the progression of kidney disease. Associations between NLRP3 inflammasome activation and oxidative stress suggest their role in the pathophysiology of PChN. The study highlights the need for further research into the mechanisms by which dialysis affects inflammation and oxidative stress in patients with kidney disease.

Main concerns:

My biggest concern is comparing groups with different numbers of outcomes (here 32, 50 and 8). Statistical analysis of data from groups with different numbers of outcomes (i.e. different sample sizes) carries significant risks and limitations that can lead to erroneous conclusions.

In smaller samples, statistics (mean, median, standard deviation, IQR) are more unstable, meaning that they can change significantly with the addition/removal of a few observations. A group with a larger number of results has greater statistical power. A group with a small number of results may produce false-negative results (failure to detect a statistical difference despite its existence). Here, the authors were very lucky because differences were found.

With 50 scores, the median is stable and represents the middle value well. It is not very sensitive to single outliers. With 8 results, the median is the average of the 4th and 5th values after sorting - it changes easily. It is therefore more susceptible to fluctuations and less representative. With 50 results, the 1st and 3rd quartiles (25% and 75% of the distribution) can be fairly accurately determined and the IQR reflects well the spread of typical values (without outliers). With 8 results the quartiles are not precise, the IQR may be falsified by single atypical values.

Caution should be taken when comparing groups with different numbers of scores. The authors rightly reported the median and IQR and indicated the sample size (n).

The question is whether tests not requiring equal numbers (e.g. non-parametric tests) were used? If not then this should be corrected in the paper.

The authors have rightly taken steps to normalise the data, so to speak, using the HSD method. However, the HSD method takes into account the size of the groups indirectly, but does not equalise them or “correct” the number of observations in a statistical sense. The formula for HSD includes, among other things, the average standard error (standard error), which depends on the sample size. Was a weighted standard error used for groups with unequal numbers? If not then this should be corrected.

It would be much better to consider data normalisation or bootstrapping averaging. This approach can be explained as follows. If one group has very little data (here 8), differences between it and a large group (here 50) can be difficult to detect, even if they are real. This is where the authors had a lot of luck.

Small groups have a larger standard error → larger HSD threshold → more difficult to demonstrate significance. In addition, ANOVA/HSD assumptions may be violated because they assume homogeneity of variance (homoskedasticity). Different counts + different variances = weakened reliability of results.

Even after applying the Tukey-Kramer HSD (as indicated in lines 269-270), results from very small groups are still not very stable and may lead to misleading conclusions.

The authors also report an extremely low correlation between the mRNA level of NLRP3 and MDA plasma levels .

Reported r=0.26 gives r2=0.068, indicating that it explains only 6.8% of the variance, which, considering the size of the sample group, may give the result equivalent to less than one sample (patient)!

 I propose to remove this from the Abstract and instead explain the statistical aspects linking this result and my queries above. It is advisable to include an additional chapter ‘Limitations of the study’ which would increase the quality of the paper considerably.

The results shown in Fig 3 had a very large scatter so logarithmic scales were used (a typical trick). This further confirms the above considerations.

Other issues:

Fig.1 does not add anything as everything is explained in the Materials and Methods section. It can be removed or details of the exclusion of patients, if any, can be added.

Table 1 is a little confusing because in the round brackets in column 1 the units of the measured quantities are given and in the next column it is IQR, although I do not understand what the case is for BMI Looks like (S.D.). Please put the units in square brackets and explain what they are.

Figs 2-4 are unreadable/blurred. What does Rho mean in Fig. 4?

Line 236 Elisa kit give the city and country of the producer.

Lines 241-242 about weight and height should be shifted to chapter Study design and patients.

Format Author contributions according to Journal standards.

Format references according to IJMS style.

The report on the similarity of texts to other previously published papers is also worrying (Percent match: 44%, iThenticate report).

Author Response

Reviewer 2

My biggest concern is comparing groups with different numbers of outcomes (here 32, 50 and 8). Statistical analysis of data from groups with different numbers of outcomes (i.e. different sample sizes) carries significant risks and limitations that can lead to erroneous conclusions.

Response: Thank you for this critical comment. We fully agree that the peritoneal dialysis (PD) group represents a small sample size (n = 8), which inevitably limits statistical power. However, it is essential to note that this reflects the actual distribution of dialysis modalities in Brazil and globally. According to the Brazilian Society of Nephrology, only approximately 4.7% of dialysis patients are on PD. In comparison, hemodialysis accounts for over 90%. International data also confirm that the prevalence of PD is around 7% worldwide. Therefore, recruiting larger numbers of PD patients within a single-center study is highly challenging. Nonetheless, we acknowledge this limitation in our manuscript and agree that the conclusions regarding the PD group should be interpreted with caution. We have also emphasized this point in the discussion section. Additionally, we employed robust statistical methods to minimize the impact of the small sample size as much as possible.

In smaller samples, statistics (mean, median, standard deviation, IQR) are more unstable, meaning that they can change significantly with the addition/removal of a few observations. A group with a larger number of results has greater statistical power. A group with a small number of results may produce false-negative results (failure to detect a statistical difference despite its existence). Here, the authors were very lucky because differences were found.

Response: We appreciate the reviewer’s comment and fully agree that smaller sample sizes can increase the risk of both Type I and Type II errors, and that statistical estimates (mean, median, standard deviation, IQR) are less stable when the number of observations is limited. In our study, we acknowledge this limitation and have explicitly addressed it in the revised manuscript (Discussion section, lines 230-234). Despite the small sample size, the observed differences reached statistical significance with appropriate nonparametric tests. We fully recognize that replication in larger samples will be crucial to confirm the findings and enhance statistical power and generalizability. We have now added text to clarify this point in the Discussion:

“Although the sample size was limited, appropriate statistical analyses were applied to reduce the influence of non-normal distributions and outliers. However, the statistically significant differences observed should be interpreted with caution, and replication in larger cohorts is warranted to confirm these results and strengthen the evidence base.”

With 50 scores, the median is stable and represents the middle value well. It is not very sensitive to single outliers. With 8 results, the median is the average of the 4th and 5th values after sorting - it changes easily. It is therefore more susceptible to fluctuations and less representative. With 50 results, the 1st and 3rd quartiles (25% and 75% of the distribution) can be fairly accurately determined and the IQR reflects well the spread of typical values (without outliers). With 8 results the quartiles are not precise, the IQR may be falsified by single atypical values.

Response: We appreciate the reviewer’s detailed comments regarding the interpretability of medians and interquartile ranges in small samples. We fully agree that in groups with limited observations (e.g., n = 8), the median and quartiles are more sensitive to fluctuations and less stable estimators of central tendency and dispersion compared to larger samples. In the present study, the use of nonparametric statistics was chosen precisely because of these sample size differences and the potential for non-normality and outliers. In addition, each data point was carefully reviewed to identify potential extreme values, and none were excluded. We acknowledge this limitation in our manuscript. Thank you for pointing out this critical methodological consideration, which we have emphasized in the revised manuscript.

Caution should be taken when comparing groups with different numbers of scores. The authors rightly reported the median and IQR and indicated the sample size (n).

Response: We appreciate the reviewer’s thoughtful observation. We agree that comparisons between groups with markedly different sample sizes require careful interpretation, as the relative instability of summary statistics in smaller samples necessitates such caution.

The question is whether tests not requiring equal numbers (e.g. non-parametric tests) were used? If not then this should be corrected in the paper.

Response: We thank the reviewer for bringing this critical point to our attention. Yes, non-parametric tests that do not require equal group sizes were applied when comparing continuous variables among the groups. Specifically, the Kruskal–Wallis test was used for continuous variables, as stated in the Statistical Analysis section. For categorical variables, the chi-squared test was used, which also tolerates unequal sample sizes. In addition, multivariable linear models were adjusted for relevant confounding variables, and estimated marginal means were compared using appropriate contrasts, with p-values corrected for multiple comparisons by the Tukey HSD method. We have revised the Statistical Analysis section to clarify this explicitly (lines 291-292) and to reinforce that non-parametric methods were used to address potential violations of normality and unequal group sizes.

Thank you for allowing us to clarify this methodological aspect.

The authors have rightly taken steps to normalise the data, so to speak, using the HSD method. However, the HSD method takes into account the size of the groups indirectly, but does not equalise them or “correct” the number of observations in a statistical sense. The formula for HSD includes, among other things, the average standard error (standard error), which depends on the sample size. Was a weighted standard error used for groups with unequal numbers? If not then this should be corrected.

Response: We appreciate the reviewer’s important observation. We confirm that the Tukey HSD procedure applied in our analyses was implemented using R software, which automatically estimates the standard error weighted according to the sample size of each group when performing multiple comparisons. Specifically, the estimated marginal means and pairwise contrasts were computed using the emmeans package and the contrast function with Tukey adjustment. This approach accounts for the unequal group sizes by applying the appropriate variance and sample-size weighting in the standard error calculation.

Thank you for the opportunity to clarify this methodological point.

It would be much better to consider data normalization or bootstrapping averaging. This approach can be explained as follows. If one group has very little data (here 8), differences between it and a large group (here 50) can be difficult to detect, even if they are real. This is where the authors had a lot of luck.

Response: We appreciate this thoughtful suggestion regarding the use of bootstrapping or data normalization to enhance the robustness of statistical inference in contexts with unequal group sizes. While we fully agree that resampling approaches, such as bootstrapping, can help address the instability of estimates in small samples, particularly when comparing them with much larger groups, our primary aim was to apply widely used non-parametric tests (Kruskal–Wallis) and conservative multiple comparison corrections to minimize potential bias.

We recognize that bootstrapped confidence intervals or alternative resampling methods could provide additional information on the uncertainty of estimates. We have now noted this as a limitation and as a recommendation for further studies in the Discussion section. Thank you for highlighting this essential methodological consideration.

Small groups have a larger standard error → larger HSD threshold → more difficult to demonstrate significance. In addition, ANOVA/HSD assumptions may be violated because they assume homogeneity of variance (homoskedasticity). Different counts + different variances = weakened reliability of results.

Response: We thank the reviewer for this precise and essential observation. We agree that small sample sizes are associated with larger standard errors and, consequently, higher Tukey HSD thresholds, making it more challenging to detect statistically significant differences. Additionally, we acknowledge that ANOVA and HSD procedures assume homogeneity of variance across groups. For continuous variables, the Kruskal–Wallis test was used as the primary method of group comparison, as it does not assume homoscedasticity and is more robust when variances and sample sizes differ. Furthermore, estimated marginal means and contrasts from the linear models were interpreted with caution, and we have explicitly acknowledged in the Discussion that differences in group sizes and variances may weaken the reliability and generalizability of the results.

Even after applying the Tukey-Kramer HSD (as indicated in lines 269-270), results from very small groups are still not very stable and may lead to misleading conclusions.

Response: We thank the reviewer for reemphasizing this critical point. We fully agree that, even after applying the Tukey-Kramer HSD adjustment, statistical estimates derived from tiny groups (such as the peritoneal dialysis subgroup, n = 8) remain less stable and should be interpreted with caution. This limitation has been acknowledged in the revised Discussion section, where we state that results for smaller groups may be prone to higher variability and reduced generalizability. Thank you for underscoring this methodological consideration.

The authors also report an extremely low correlation between the mRNA level of NLRP3 and MDA plasma levels.

Response: We appreciate the reviewer’s observation. Indeed, our results showed a very low correlation between NLRP3 mRNA levels and MDA plasma concentrations. We agree that this suggests that inflammasome activation at the transcript level may not be directly coupled with systemic oxidative stress, as measured by MDA, or that additional regulatory mechanisms and post-transcriptional modifications influence the relationship.

Reported r=0.26 gives r2=0.068, indicating that it explains only 6.8% of the variance, which, considering the size of the sample group, may give the result equivalent to less than one sample (patient)! I propose to remove this from the Abstract and instead explain the statistical aspects linking this result and my queries above. It is advisable to include an additional chapter ‘Limitations of the study’ which would increase the quality of the paper considerably.

Response: We appreciate this valuable suggestion. We agree that the reported correlation coefficient (r = 0.26), corresponding to approximately 6.8% of the variance explained, indicates a weak association that should be interpreted with caution, particularly given the sample size. Following your recommendation, we have removed this correlation result from the Abstract to avoid overemphasizing its relevance and added further clarification in the Results and Discussion sections regarding the limited explanatory power of this association. Included a new subsection entitled “Limitations of the study,” where we explicitly discuss this point along with other methodological limitations (small sample size, unequal group sizes, potential residual confounding, etc.). Thank you for this constructive advice to improve the transparency and rigor of the manuscript.

The results shown in Fig 3 had a very large scatter so logarithmic scales were used (a typical trick). This further confirms the above considerations.

Response: We appreciate the reviewer’s observation. Indeed, the data presented in Fig. 3 showed substantial scatter, which led us to use logarithmic scales to visualize the distribution and relationships between variables better. We agree that this reinforces the need for cautious interpretation, as the variability further underscores the exploratory nature of the findings. Thank you for pointing this out.

Other issues:

Fig.1 does not add anything as everything is explained in the Materials and Methods section. It can be removed or details of the exclusion of patients, if any, can be added.

Response: Thank you for this comment. We agree that the information contained in the Consort flowchart is already explained in the study methodology. Therefore, we agree to remove the figure from the study.

Table 1 is a little confusing because in the round brackets in column 1 the units of the measured quantities are given and in the next column it is IQR, although I do not understand what the case is for BMI Looks like (S.D.). Please put the units in square brackets and explain what they are.

Response: We appreciate this comment and understand the concern regarding possible confusion between the units and the statistical measures presented in Table 1. We want to clarify that all continuous variables, including BMI, are presented consistently as median (interquartile range), as stated in the table legend. The use of parentheses for the units in the first column follows conventional formatting; however, since the legend explicitly indicates that all values are reported as medians (with interquartile ranges), this avoids any potential misinterpretation. Nevertheless, to improve clarity, we have carefully reviewed the formatting and ensured that the presentation is as straightforward as possible.

Figs 2-4 are unreadable/blurred. What does Rho mean in Fig. 4?

Response: Regarding the quality of Figures 2–4, we agree that the resolution in the submitted version was insufficient. We have replaced them with high-resolution images to ensure readability. Indeed, the use of “Rho” in Figure 4 was an oversight on our part. The correlation shown corresponds to the Pearson correlation coefficient, not Spearman’s rho. We have corrected the figure accordingly, replacing “Rho” with “r” to indicate the Pearson correlation coefficient. Additionally, we have updated the figure legend to specify that Pearson’s r is reported clearly.

Thank you for allowing us to clarify this point.

Line 236 Elisa kit gives the city and country of the producer.

Response: Thank you for this comment. We have corrected it.

Lines 241-242 about weight and height should be shifted to chapter Study design and patients.

Response: Thank you for your suggestion. We have moved it.

Format Author contributions according to Journal standards.

Format references according to IJMS style.

Response: Thank you for this suggestion. We have corrected them.

The report on the similarity of texts to other previously published papers is also worrying (Percent match: 44%, iThenticate report).

Response: We thank the reviewer for this vital observation regarding text similarity. We want to emphasize that the authors prepared all content in the manuscript, and it is original. We acknowledge that specific methodological descriptions and standardized expressions may be similar to those found in previous publications on related topics. To address this concern more effectively, we request that you share either the iThenticate report or specific examples of matched content with us. This will enable us to review any overlapping sections in detail and, if necessary, rephrase or appropriately cite any content to ensure full compliance with publication standards. Thank you for your assistance in clarifying this point.

Once again, we thank the Reviewer for their constructive criticism, which has enabled us to improve the manuscript to the point where we hope it can be accepted for publication in IJMS.

Denise Mafra, on behalf of the rest of the co-authors.
